# Surgical Site Infection after Primary Open Surgery for Laryngeal Cancer in a Tertiary Hospital in Belgrade, Serbia: A 10-Year Prospective Cohort Study

**DOI:** 10.3390/antibiotics13100918

**Published:** 2024-09-25

**Authors:** Jelena Sotirović, Nemanja Rančić, Ljubomir Pavićević, Nenad Baletić, Aleksandar Dimić, Ognjen Čukić, Aleksandar Perić, Milanko Milojević, Nenad Ljubenović, Darko Milošević, Vesna Šuljagić

**Affiliations:** 1Clinic for Otorhinolaryngology, Military Medical Academy, 11000 Belgrade, Serbia; ljpavicevic@gmail.com (L.P.); nenadbaletic@yahoo.com (N.B.); dimicdraleksandar@gmail.com (A.D.); ognjen.cukic.bg@gmail.com (O.Č.); aleksandarperic1971@gmail.com (A.P.); milankom@mts.rs (M.M.); 2Medical Faculty, Military Medical Academy, University of Defence, 11000 Belgrade, Serbia; nece84@hotmail.com (N.R.); darko9979@gmail.com (D.M.); suljagicv@gmail.com (V.Š.); 3Center for Clinical Pharmacology, Military Medical Academy, 11000 Belgrade, Serbia; 4Institute of Epidemiology, Military Medical Academy, 11000 Belgrade, Serbia; nljubenovic90@gmail.com; 5Department of Healthcare-Related Infection Control, Military Medical Academy, 11000 Belgrade, Serbia

**Keywords:** surgical site infection, laryngeal cancer, risk factors, antibiotic prophylaxis, Gram-negative bacteria

## Abstract

Background: Surgical site infection (SSI) in laryngeal cancer (LC) patients significantly increases morbidity and may postpone adjuvant therapy. Additionally, SSI can prolong hospitalization, thus representing a burden for the healthcare system. Most of the published studies refer to SSI after salvage laryngectomy. Methods: The present prospective cohort study aimed to clarify the incidence and factors associated with SSI in patients after primary open surgery for LC. Through regular hospital surveillance of patients who underwent primary partial or total laryngectomy, we gathered 24 putative factors and identified SSI from 2013 to 2022. Patients with SSI were compared with patients without SSI. Results: SSI was observed in 21 (6.6%) of 319 patients. ULRA showed that the occurrence of SSI was significantly associated with the American Society of Anesthesiologists (ASA) score, other postoperative healthcare-associated (HAI) infection, T classification, N classification, advanced clinical stage (III–IV), length of stay (LOS), duration of drainage, and the National Healthcare Safety Network (NHSN) risk index. Multivariate logistic regression analysis identified two independent factors associated with SSI occurring in these patients: duration of drainage (RR (relative risk) 1.593; 95% CI 1.159–2.189; *p* = 0.004) and LOS (RR: 1.074; 95% CI: 1.037–1.112; *p* < 0.001). Conclusions: Our study provided insight into the burden of SSI in LC patients, highlighting several priority areas and targets for quality improvement.

## 1. Introduction

The most common consequences of surgical site infection (SSI) in cancer patients are prolonged hospitalization and possible delayed adjuvant treatment, which, altogether, may alter the oncologic results, as well as represent a significant burden for the health care system [1,2,3]. Patients undergoing laryngeal cancer surgery are at risk of developing SSI because of the clean–contaminated surgical field. Exposition to bacterial contamination and the need to recreate the mucosal barrier of the pharynx and larynx increase the likelihood of SSIs [4].

In published studies, numerous risk factors for SSI in head and neck cancer (HNC) patients have been reported, such as alcohol consumption, diabetes mellitus, tumor (T) classification, neck dissection, reconstructive procedure, chemoradiotherapy, advanced stages, tracheotomy, duration of surgery, preoperative albumin, and the nodal (N) stage [5,6,7,8]. The frequency of SSI following HNC surgery has been reported to range from 3% to 41% [9]. Previous studies on SSI following laryngeal surgery are scarce, and most of them refer to salvage laryngectomy. Preference for chemoradiotherapy (CRT) as the primary therapeutic approach for laryngeal cancer maintains the function of the larynx yet may represent an additional potential risk factor for SSI in case of salvage laryngectomy, causing fibrosis and perivascular necrosis [5].

This present study aimed to clarify the incidence and risk factors associated with SSI in patients after primary open surgery for laryngeal cancer (LC). Identifying the factors associated with SSI prior to cancer treatment may help decision making; therefore improving the management of laryngeal cancer patients and perioperative care.

## 2. Results

A total of 319 surgical procedures were evaluated (Table 1). The mean age of the patients was 62.93 ± 8.97 years. Of these, 289 (90.6%) were men and 30 (9.4%) were women. The mean LOS was 22.50 ± 12.53 days.

SSI was observed in 21 (6.6%) of 319 patients. All participants were smokers. The type of SSI in our cohort is presented in Table 2.

The characteristics of the patients and SSI-related factors according to univariate logistic regression analysis (ULRA) are shown in Table 1. ULRA showed that the occurrence of SSI was significantly associated with the following variables: American Society of Anesthesiologists (ASA) score, other healthcare-associated infections (HAIs), T classification, N classification, stage III + IV, length of hospital stay (LOS), duration of drainage (keeping the drain in place), and the National Healthcare Safety Network (NHSN) risk index. Age, diabetes mellitus, gender, body mass index (BMI), in-hospital mortality, intensive care unit (ICU) admission, length of stay in ICU, preoperative length of stay, usage of central vascular catheters (CVCs), usage of urinary catheters (UCs), type of antibiotic used in prophylaxis as the first antibiotic (cephalosporins first generation, cephalosporins second generation, cephalosporins third generation, quinolones, penicillin, macrolides, sulfonamides), laryngeal surgery (partial or total laryngectomy), duration of operation, neck dissection, and drainage were not found to be associated with SSI in ULRA.

Multivariate logistic regression analysis (MLRA) identified two independent factors associated with SSI occurring in these patients: duration of drainage (RR 1.593; 95% CI 1.159–2.189; *p* = 0.004) and LOS (RR: 1.074; 95% CI: 1.037–1.112; *p* ˂ 0.001) (Table 1).

Microorganisms were isolated in 15 (71.4%) out of 21 recorded SSI. Among these, three (14.3%) SSI were polymicrobial. Bacteria from the order *Enterobacterales* (*Escherichia coli*, *Klebsiella* spp., and *Proteus* spp.) were the most frequently isolated microorganisms causing 6 (40.0%) of laboratory-confirmed SSIs, followed by *Pseudomonas aeruginosa*, which were isolated in 4 (26.7%) of laboratory-confirmed SSIs. *Coagulase-negative Staphylococcus* was the third most frequent causative agent of SSI, causing 3 (20.0%) of the laboratory-confirmed SSIs, followed by *Staphylococcus aureus* causing 2 (13.3%) of the laboratory-confirmed SSIs. Apart from these, other identified species were: *Acinetobacter* spp., *Enterococcus* spp., *Streptococcus* spp., and *Candida* spp. isolated from one SSI sample each (Figure 1).

## 3. Discussion

The most frequent type of laryngeal cancer is squamous cell carcinoma [10]. In the last decade, laryngeal cancers have increased in incidence worldwide, especially in countries with a lower sociodemographic index, where smoking and alcohol consumption habits are highly widespread [11]. In Serbia, a country with a high–middle sociodemographic index, laryngeal cancer is the seventh among the ten most frequently registered cancers in men, behind lung and bronchus, colon and rectum, prostate, bladder, pancreas, and stomach cancers. In the total number of newly reported cases of malignant diseases in men, they participate with 3.2% [12].

SSI is one of the inevitable complications of laryngeal cancer surgery which increases morbidity and health care costs, as we reported in our previous investigation during the period 2006–2011 [13]. The specificity of laryngeal cancer surgery is a clean-contaminated surgical field. Surgical wounds encompass the mucosa of the respiratory and digestive tract, and previous studies show that the rate of SSI in clean–contaminated head and neck surgery can be more than 80% without antibiotic prophylaxis [9]. Compared to head and neck surgery, patients with head and neck cancer (HNC) are at a higher risk for developing SSIs [14]. Moreover, SSI may even influence the perioperative mortality in HNC patients [15].

In the case of laryngeal cancer, surgery treatment options include endoscopic approaches, primary open surgery, and salvage surgery. In the past decades, new treatment modalities were introduced for laryngeal cancer that preserve the morphology and functionality of the larynx. Organ preservation in laryngeal cancer focuses on treating the disease effectively while maintaining the patient’s ability to speak and swallow, thereby preserving their quality of life. This strategy typically involves a multimodal approach combining chemotherapy and radiation therapy. Nevertheless, previous radiotherapy leads to reduced collagen deposition and angiogenesis, soft tissue fibrosis, and perivascular necrosis, which thus may obscure the surgical plane and impede tissue healing [16,17].

According to the published literature, complications after salvage laryngectomy are high [5,8,18,19]. In one study, the overall complication rate in patients with salvage total laryngectomy after chemotherapy followed by radiation therapy, radiation therapy alone, or concomitant CRT was up to 59% [20]. Similarly, Ganly et al. reported that 32% of patients with salvage total laryngectomy after CRT developed a pharyngocutaneous fistula [21]. Gan et al. found a 2.9-fold increase in the odds of SSI in patients with HNC in the preoperative radiotherapy group compared to the non-preoperative radiotherapy group [6]. Furthermore, Suzuki et al. showed that the SSI rate was higher in patients with salvage laryngectomy after cetuximab-based bioradiotherapy compared with patients treated with platinum base chemoradiotherapy [19]. Compared to salvage laryngectomy, the incidence of SSI in our cohort of patients with primary open surgery is low (6.6%). In terms of SSI, it is a clear advantage of primary over salvage surgery. In our opinion, primary surgery should be considered an important modality of treatment among the various therapies for LC.

Preoperative assessment and evaluation are critical for success and for minimizing complications. An important aspect of preventing SSI is to pay careful attention to surgical cases with high-risk factors. To reduce the rate of SSI, patients, hospitalization, and surgery-related measures are to be combined. This study identifies the duration of hospital stay, incidence, and risk factors in primary open laryngeal surgery to tailor prevention strategies to individual risk profiles.

ASA score was found to be a significant risk factor for postoperative infection in HNC surgery, as reported in previous studies [6,7,9]. Our ULRA results also demonstrated that patients with a higher ASA score were more likely to have a SSI (*p* ˂ 0.001). The ASA score provides a standardized measure of a patient’s overall health status and physiological condition before undergoing surgery with higher ASA scores indicating significant comorbidities. By integrating the ASA score into preoperative assessments, surgeons can better anticipate and manage risks, including SSI, ultimately leading to improved patient safety and surgical outcomes. Thorough preoperative evaluations are mandatory to assess the patient’s health comprehensively and address any issues that could be a potential risk for surgical site infections. Strategies to improve a patient’s ASA score include managing chronic diseases such as diabetes, hypertension, and heart disease through medication and lifestyle changes by optimizing blood glucose levels, cholesterol levels, and other metabolic markers. In preparation for surgery, surgeons should encourage physical activity to improve cardiovascular fitness and overall strength, suggest rehabilitation for any physical impairments or functional limitations that may affect the patient’s health status, and address any malnutrition or deficiencies. This may include other specialists to address complex medical issues or to improve the management of specific health conditions. By addressing these areas, healthcare professionals can help improve a patient’s overall health status, potentially leading to a better ASA score and reduced SSI risk.

NHSN risk adjustment that calculates differences in patient case mix is critical to allow for more meaningful comparisons between surgeons, hospitals, or operative procedures. Using the traditional NHSN risk index [22], we registered 52.4% of patients with SSI compared to 20.1% of patients without SSI within risk 2. Our ULRA identified that the overall risk of SSI was almost doubled among patients at NHSN risk 2 (RR: 0.262; 95%CI: 0.097–0.707; *p* = 0.008) but did not retain significance as independent factors in MLRA (*p* = 0.392). A new risk assessment model applied to neck surgery has also been shown to be useful but we did not use it with our patients [23].

More than 90% of our patients with SSI and 63.4% of patients without SSI were in the stage of malignant disease designated as T3 and T4, which ULRA assessed as statistically significant. Furthermore, we observed a 4.4-fold increase in the SSI rate in patients with advanced stages (III–IV) in ULRA. Our results are in conclusion with other studies [5,7,24]. The Union for International Cancer Control (UICC) staging system for laryngeal cancer categorizes the disease based on the extent of the primary tumor (T), regional lymph node involvement (N), and the presence of distant metastasis (M). T stages range from a small, localized tumor (T1) to a large (T4) tumor. N stages range from N0, with no regional lymph node involvement, to N3, indicating extensive nodal spread. The stage of laryngeal cancer reflects the extent of the disease and often involves more extensive surgical procedures, possibly associated with a greater burden on the body’s immune system. Advanced cancer can also lead to malnutrition or altered nutritional status, weakening the body’s ability to mount an effective immune response. The cancer itself can directly affect the immune system or create conditions that make it easier for infections to take hold. These factors collectively increase the susceptibility of advanced cancer patients to infections, making management and preventive care crucial in their treatment plans. Advanced-stage laryngeal cancer may anticipate potential complications and tailor infection prevention strategies to improve patient outcomes and minimize the risk of surgical site infections.

A nasogastric (NG) feeding tube is typically used after a supraglottic and total laryngectomy. The NG tube provides a temporary means of nutrition and hydration while the patient recovers and adapts to their new method of swallowing, or until they can resume normal oral intake. Foreign bodies, such as NG feeding tubes, may be colonized by bacteria forming a biofilm that may hinder antibiotic effectiveness in a hospital environment [25]. Although Pecorari et al. showed that the NG feeding tube is a significant factor in SSI occurrence, ULRA in our study did not reveal an association between the NG feeding tube and SSI [7].

The duration of drainage was found to be independently associated with SSI. This is rather a consequence than the risk factor for SSI development since drainage aims to drain pus or exudate from the infected wound. It is reasonable to assume that, as a foreign material, a surgical drain may also contribute to the infection. Nevertheless, we did not find an association between drainage and SSI occurrence. The study of Seneviratne et al. showed an increased rate of microbial contamination in neck drains used during procedures involving the upper aero–digestive tract and neck [26]. While the difference was not statistically significant, there was a higher incidence of normal skin flora contamination in drains from clean–contaminated compared to clean procedures. Also, the rates of pathogenic skin organisms and pathogenic oropharyngeal organisms were comparable between patients undergoing clean–contaminated and clean procedures.

MLRA identified that SSI was connected to longer LOS, while the preoperative length of stay did not differ between groups of patients with and without SSI. Prolonged hospitalization inevitably leads to higher costs [2,7]. Our data did confirm that SSI poses a great burden to the healthcare system extending the length of hospital stay in these patients. Additionally, we observed that other HAIs are significantly more frequent in patients with SSI. Longer hospitalization in patients with SSI might be a reason for more frequent postoperative pneumonia and urinary infections in our study. Patients with prolonged LOS are more exposed to potential sources of infection, such as contaminated surfaces, medical equipment, healthcare workers, and other patients [27,28].

The most frequently isolated bacteria in our study were *Enterobacterales* (*Escherichia coli*, *Klebsiella* spp., and *Proteus* spp.). Other authors also found Gram-negative bacteria as a significant factor for SSI in HNC patients [29,30]. Infection control measures for Enterobacteriales are essential to prevent infection in healthcare settings. Key measurements include regular hand hygiene, use of personal protective equipment, cleaning and disinfecting surfaces and equipment, active surveillance, antimicrobial stewardship, etc. Also, patient education is of utmost importance. It is necessary to inform patients about the importance of hygiene and infection prevention measures. A relative low incidence of *Staphylococcus aureus* 2 (11.1%) may be due to preoperative bathing with a chlorhexidine-based solution, among other things.

Keeping bacterial contamination of the surgical site to a minimum is the most important prophylaxis, and oral care before and after surgery was proven to reduce postoperative infections in patients undergoing HNC surgery [31]. In terms of prophylaxis, some of the most important questions are should perioperative antibiotics be used, which antimicrobial agent, and for how long. According to previous studies, without prophylactic administration of antibiotics, the incidence of SSI in clean-contaminated head and neck cancer surgery can be up to 87% [9]. The reason for the high incidence of SSI in laryngeal surgery is the contamination of surgical wounds by oropharyngeal secretions. Considering that laryngeal surgery is a clean-contaminated surgical field, antibiotic prophylaxis is recommended [32].

The right choice of antibiotic prophylaxis is based on the fact that surgical wounds in head and neck patients show colonization with Gram-positive, enteric Gram-negative, and anaerobic bacteria [33]. For this reason, recommendations include penicillin, cephalosporin, metronidazole, and aminoglycosides [34]. The latter is frequently advocated as a second drug in case of penicillin allergy when clindamycin is prescribed. Clindamycin was found to be ineffective as a single antimicrobial agent in SSI prevention in numerous studies [4,34,35]. This is especially true for Gram-negative coverage. It is worth mentioning that penicillin allergies were found to be over-diagnosed and that the true drug allergy should be evaluated by an immunologist/allergist by skin-prick or serum IgE test [36]. Our study demonstrated that although cefalosporins are predominant as a prophylactic antibiotic used in our cohort of patients, there was no association between SSI and the administration of other antibiotics such as quinolones, penicillin, macrolides, and sulfonamides (Table 1).

The duration of prophylactic antibiotic agents is based on the established guidelines [37]. There is still wide variation in the implementation of recommended regimens. Presumably, personal experience plays an important role in decision making [24,38]. Because of the risks of SSI and its possible consequences on treatment outcomes, surgeons overuse antibiotics for fear of infectious complications and are unaware of the adverse effects of antibiotics, such as resistance to microorganisms and diarrhea associated with *Clostridioides difficile* [39,40].

According to the global guideline for the prevention of SSI published by WHO, the antibiotic prophylaxis is defined as the administration of a single dose of the effective antimicrobial agent 120 min before surgical incision [41]. Furthermore, the Centers for Disease Control and Prevention (CDC) recommended that additional prophylactic antimicrobial agent doses should not be administered after the surgical incision in clean and clean–contaminated procedures, even in the presence of a drain [42]. Contrary to the recommendations, repeated studies of the prevalence of HAI in hospitals in Serbia showed that more than 70 surgical prophylaxis courses were prescribed for more than 1 day in both years of investigation, 2017 and 2022 [43,44].

In the available literature, the terms “short” and “long” duration significantly differ among authors. Short duration ranges from 1–3 to 1–7 days [4,24,34]. In a recently published study, perioperative antibiotic prophylaxis was given for 7 to 10 days postoperatively [45]. Busch et al. concluded that prophylactic antibiotic treatment should not exceed 7 days in patients without increased risk for surgical wound infections, while Veve et al. stated that the duration of 1–5 days seems to have the highest level of evidence [24,33].

Data on the causative agents of SSIs in laryngectomy patients are scarce. In our previous study, we showed a similar profile of causative agents of SSIs with the domination of Gram-negative bacteria, which is not surprising because it is the same hospital where healthcare workers work with similar habits in prevention, as well as prescribing antibiotic prophylaxis [13]. Such a pathogen landscape can only be compared to other hospitals in Serbia that identified identical patterns of causative agents of SSI in their latest surveys of the prevalence of HAI guided by the European Center for Disease Prevention and Control between 2017 and 2022 [44,46].

SSI with a negative culture report in routine diagnosis is not rare and depends on several circumstances, the most important of which are the type of surgery, the quality of the sample taken for diagnosis, and the quality of microbiological diagnostics [47,48,49]. In our report, 28.6% of patients with SSIs were culture-negative and challenged the selection of appropriate treatment options and prognosis prediction. Additional communication with the medical microbiologist is strongly recommended to ensure the optimal recovery of organisms in these cases. Anaerobic pathogens of infections should not be forgotten and, in this sense, the diagnosis of SSI in our patients should be improved.

The strength of this 10-year-long study is its prospective cohort design. Also, it is a real-life study. The limitation of this study was that we did not analyze how alcohol consumption, blood loss, hemoglobin level, albumin level, and intraoperative blood loss influenced SSI in our patients. The analysis of these factors could have enhanced the relevance of our results. Another limitation is that this study was conducted at a single center, which limits the generalizability of the results.

## 4. Materials and Methods

This prospective cohort study was performed from 1 January 2013, to 31 December 2022, in the Clinic for otorhinolaryngology, Military Medical Academy (MMA), Belgrade. A total of 319 consecutive patients who underwent open surgical treatment as an elective procedure for laryngeal squamous cell carcinoma as curative therapy were enrolled. All patients used chlorhexidine-based solution for preoperative showers. Surgical procedures included partial or total laryngectomy with or without partial pharyngectomy and neck dissection if required. All patients had tracheostomy postoperatively and received antibiotic prophylaxis. Antibiotic prophylaxis was administered on the day of surgery and 6 days after surgery, consisting of first-generation cephalosporin/second-generation cephalosporin/third-generation cephalosporin or penicillin as the first antibiotic, with the addition of metronidazole and amikacin as the second and third antibiotics. In those with penicillin allergy quinolones, penicillin, macrolides, or sulfonamides instead of cephalosporins were given. Patients with LSCC treated with an endoscopic approach and other malignant tumors of the larynx were excluded from this study.

All patients were assessed before the operation by an anesthesiologist for ASA physical status score. After surgery, these patients were followed, and the outcome—whether they developed SSI—was identified. Surgical wounds were observed daily until patient discharge. SSI was defined according to the Centers for Disease Control (CDC) and the NHSN surveillance definitions and included superficial incisional, deep incisional, and organ/space infection [50]. The personnel of the Department of Infection Control, MMA performed surveillances for HAI. An infectious disease specialist was included in the treatment of patients with SSIs and other postoperative infections if needed. The microbiological testing of wound swabs was performed at MMA’s Institute of Medical Microbiology. Also, we identified patients who were readmitted due to SSI that developed post-discharge.

We gathered 24 putative factors in patients with and without SSI. Patient characteristics included age, gender, diabetes mellitus, body mass index (BMI), ASA score, other HAI, and in-hospital mortality. Also, we included T classification, N classification, and clinical staging according to the UICC, eighth edition [51]. Parameters related to hospitalization consisted of ICU hospitalization, length of ICU stay, preoperative length of stay, usage of CVC, usage of UC, LOS, and the type of antibiotic used in prophylaxis as the first antibiotic (cephalosporins first generation, cephalosporins second generation, cephalosporins third generation, quinolones, penicillin, macrolides, sulfonamides). Surgery procedure characteristics included the type of laryngectomy (partial or total), neck dissection, NG feeding tube, drainage, duration of drainage, duration of operation, and NHSN risk. The NHSN risk is calculated on the data related to the operation and ranges from 0 to 3. Each of the three variables (contaminated or dirty surgical wound, ASA scores greater than 2, and the duration of surgery greater than the 75th percentile for a specific group of surgical procedures) represents 1 point [22,23].

Statistical analysis was performed using IBM SPSS 26.0 software (Chicago, IL, USA). Data are presented as mean ± SD, range (minimum, maximum), and count (%). The univariate relationship between each variable and SSI was analyzed with univariate and multivariate logistic regression. Only variables with a *p*-value less than 0.05 in ULRA and variables that could be potential risk factors for infection were entered in a MLRA.

## 5. Conclusions

As a single-center study, this paper offers insight into factors associated with SSI and the duration of hospitalization in LC patients treated with primary open surgery in a tertiary hospital. In our opinion, given the low SSI incidence, surgery remains an important primary treatment modality for LC. The length of hospital stay and duration of drainage were independently associated with the occurrence of SSI in our patients, showing that SSI poses a great burden to the healthcare system. Future directions to reduce SSI should focus on identifying high-risk patients with a tailored pharmacologic approach, as well as more effective collaboration between head and neck cancer surgeons and infection prevention and control teams.

## Figures and Tables

**Figure 1 antibiotics-13-00918-f001:**
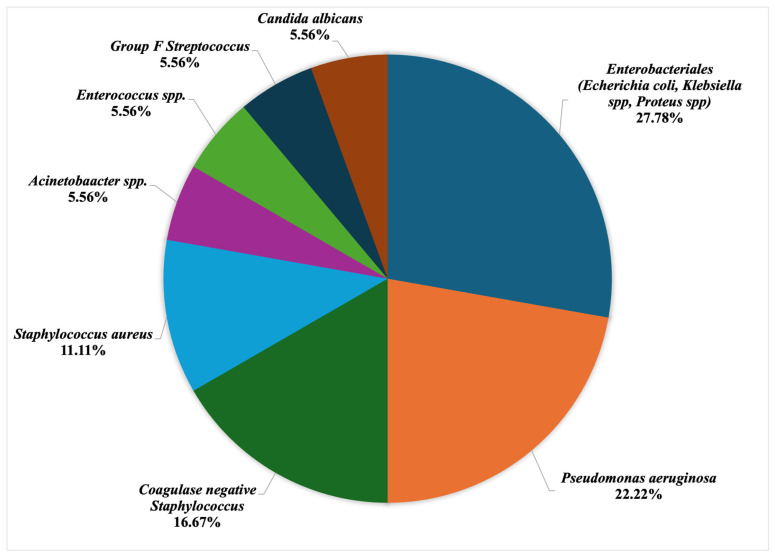
Distribution of isolated microorganisms by percentage.

**Table 1 antibiotics-13-00918-t001:** Potential risk factors for acquisition of surgical site infections (SSI) in patients after primary open surgery for laryngeal cancer: results of univariate logistic regression analysis and multivariate regression analysis.

Patients Characteristics	Patients with SSI n = 21 (%)	Patients without SSI n = 298 (%)	ULRA OR (95% CI)	*p*	MLRA OR (95% CI)	*p*
Male, n (%)	19 (90.5)	270 (90.6)	0.985 (0.218–4.451)	0.985		
DM, n (%)	3 (14.3)	38 (12.8)	1.140 (0.321–4.056)	0.839		
Age, years ± SD	63.9 ± 8.6	62.7 ± 9.7	1.014 (0.967–1.063)	0.564		
BMI ± SD	27.4 ± 2.9	26.3 ± 4.0	1.083 (0.956–1.226)	0.213		
ASA score			0.167 (0.065–0.429)	˂0.001		
ASA score 1, n (%)	0 (0.0)	5 (1.7)				0.253
ASA score 2, n (%)	15 (71.4)	243 (81.5)				0.999
ASA score 3, n (%)	6 (28.6)	50 (16.8)			0.201(0.030–1.339	0.097
HAI, n (%)	2 (9.5)	4 (1.3)	7.737 (1.332–44.954)	0.023	1.110 (0.121–10.185)	0.926
T, n (%)						
T1, n (%)	0 (0.0)	40 (13.4)	Ref.			
T2, n (%)	2 (9.5)	69 (23.2)		0.998		0.997
T3, n (%)	8 (38.1)	118 (39.6)	0.187 (0.040–0.875)	0.033		0.998
T4, n (%)	11 (52.4)	71 (23.8)	0.438 (0.168–1.140)	0.091	0.457 (0.131–1.592)	0.219
N, n (%)						
Nx, n (%)	7 (33.3)	135 (45.3)	3.857 (0.706–21.065	0.119	0.828 (0.098–6.984)	0.862
N0, n (%)	6 (28.6)	90 (30.2)	Ref.			
N1, n (%)	4 (19.0)	18 (6.0)	1.286 (0.418–3.951)	0.661	1.159 (0.281–4.787)	0.838
N2, n (%)	2 (9.5)	45 (15.1)	4.286 (1.141–16.096)	0.031	3.480 (0.578–20.972)	0.174
N3, n (%)	2 (9.5)	10 (3.4)	0.857 (0.172–4.276)	0.851	0.614 (0.082–4.598)	0.635
Stage III + IV	19 (90.5%)	203 (68.1%)	4.446 (1.015–19.477)	0.048		0.998
Stage I, n (%)	0 (0.0)	38 (12.8)				
Stage II, n (%)	2 (9.5)	57 (19.1)				
Stage III, n (%)	5 (23.8)	99 (33.2)				
Stage IV, n (%)	14 (66.7)	104 (34.9)				
In hospital mortality, n (%)	1 (4.8)	3 (1.0)	5.236 (0.681–40.230)	0.112		
Procedures during hospitalization						
ICU hospitalization, n (%)	4 (19.0)	26 (8.7)	2.462(0.771–7,861)	0.128		
Length of ICU stay, days ± SD	0.7 ± 2.4	0.1 ± 0.3	1.715(0.864–3.405)	0.123		
Preoperative LOS, days ± SD	9.1 ± 8.9	7.4 ± 7.2	1.028(0.976–1.083)	0.295		
CVC, n (%)	1 (4.8)	5 (1.7)	2.930 (0.327–26.293)	0.337		
UK, n (%)	3 (14.3)	35 (11.7)	1.252 (0.351–4.469)	0.729		
LOS, days ± SD	42.9 ± 22.1	21.1 ± 10.2	1.090 (1.057–1.125)	˂0.001	1.074 (1.037–1.112)	˂0.001
Surgical prophylaxis				0.974		
Cephalosporins first gen.	1 (4.8)	8 (2.7)				
Cephalosporins second gen.	15 (71.4)	168 (56.4)				
Cephalosporins third gen.	5 (23.8)	97 (32.6)				
Quinolones	0	3 (1.0)				
Penicillin	0	8 (2.7)				
Macrolides	0	1 (0.3)				
Sulfonamides	0	13 (4.4)				
Surgery procedure characteristics						
PL, n (%)	3 (14.3)	90 (30.2)	2.596 (0.746–9.034)	0.134		
TL, n (%)	18 (85.7)	208 (69.8)				
NG feeding tube, n (%)	20 (95.2%)	235 (78.9%)	5.362 (0.706–40.724)	0.105		
Neck dissection, n (%)	14 (66.7%)	163(54.7%)		0.286		
Drainage, n (%)	21 (100%)	289 (97.0%)		0.99		

SSI: surgical site infection; ULRA: univariate linear regression analysis; MLRA: multivariate linear regression analysis; SD: standard deviation; DM: diabetes mellitus; BMI: body mass index; ASA score: American Society of Anesthesiologists physical status classification; HAI: healthcare-associated infection; ICU: intensive care unit; LOS: length of stay; CVC: central vascular catheter; UK: urinary catheter; PL: partial laryngectomy; TL: total laryngectomy; NG: nasogastric; Ref: reference value; *p*: probability.

**Table 2 antibiotics-13-00918-t002:** Type of SSI.

Type of SSI	No of Patients	%
Superficial	3	14.28
Deep	9	42.86
Organ/space	9	42.86
Total	21	100

SSI = surgical site infection.

## Data Availability

The data sets used and/or analyzed in this present study are available from the corresponding author upon reasonable request.

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
