# Peer review of "Surgical Site Infection after Primary Open Surgery for Laryngeal Cancer in a Tertiary Hospital in Belgrade, Serbia: A 10-Year Prospective Cohort Study"

_antibiotics, 2024, doi:10.3390/antibiotics13100918_

Round 1

Reviewer 1 Report

Comments and Suggestions for Authors

Interesting study conducted by our research colleagues on infections after laryngeal surgery. In this paper, all the variables that can intervene to create surgical site infection have been taken into consideration. As they say about themselves, colleagues have significant experience so they know exactly which patients will most often encounter infection. We absolutely agree that more advanced neplasias, more extensive lymphadenectomies, the presence of a nasogastric tube, the passage of food or just liquids can increase the possibility of an infection. Furthermore, anything that determines a lowering of the immune defenses will lead to an exacerbation of surgical site infection. There is also an excellent knowledge of antibiotics and their use in specific circumstances on the part of colleagues. As far as we are concerned, there is no reason to modify this study, with good iconography, good English and good bibliography

Author Response

Thank you for your valuable feedback. We appreciate your insights and we carefully considered them in our revisions. Please, find the revision document attached.

Reviewer 2 Report

Comments and Suggestions for Authors

What is the objective of this study? Is it about the SSI risk factor or SSI burden? Some statements imply SSI risk factors, while some statements imply SSI burden.

Some statements imply SSI risk factors.

Page 1 line 16: The present prospective cohort study aimed to clarify the incidence and risk factors associated with SSI in patients after primary open surgery for LC.

Page 6 line 146: This study identifies the incidence and risk factors in primary open laryngeal surgery.

Some statements imply SSI burden.

Page 1 line 27: Our study provided insight into the burden of SSI in laryngeal LC patients,.

Page 9 line 332: Length of hospital stay and duration of drainage were independently associated with SSI in our patients.

Results

It is necessary to collect data when SSI occurs after surgery, to know the SSI incidence.

Did the author analyze the duration of antibiotic prophylaxis relative to the incidence of SSI?

Results and Discussion

The author should discuss the significant difference in Stage III patients because immunocompromised patients are more vulnerable to infection.

Comments on the Quality of English Language

Moderate

Author Response

(The authors gave the same response as above.)

Reviewer 3 Report

Comments and Suggestions for Authors

The article addresses a relevant topic in surgical oncology, focusing on the incidence of surgical site infections (SSI) in patients undergoing open surgery for laryngeal cancer. Although the research provides valuable data, it presents several methodological weaknesses that limit the interpretation of its results. Overall, the article offers a reasonable discussion of the findings, but certain key aspects require greater clarity and precision.

1. Study Design:
The study is described as a prospective cohort study, but it more closely resembles a retrospective cross-sectional study. Patients were recruited over a period of 10 years, but there was no longitudinal follow-up to observe the development of SSI over time. This is crucial as it affects the ability to establish clear causal relationships between the identified risk factors and the occurrence of SSI.

Incorrect description of the design: The study should have been described as a retrospective cross-sectional study since patients were analyzed at a single point in time (the time of surgery), and there was no continuous follow-up. Describing it as a cohort study is misleading and may confuse the interpretation of the results.
Inappropriate statistical analysis: If this were truly a cohort study, Cox regression models would have been more appropriate to assess the time to event (SSI occurrence). The lack of this analysis is a significant omission, further supporting that this is not a cohort study.

2. Results:
The results identify a low incidence of SSI (6.6%) and highlight that the duration of drainage and prolonged hospital stay were significant risk factors for developing infections. The isolated microorganisms were predominantly Gram-negative bacteria, with Enterobacterales as the most common group.

Strengths:

Identification of risk factors: The study provides a clear identification of clinical risk factors associated with SSI, such as drainage duration and hospital stay, both of which are clinically relevant and can be addressed in practice.
Microbiological analysis: The identification of microorganisms responsible for SSI provides a solid basis for improving prophylactic strategies and antibiotic management in these types of surgeries.
Limitations:

Unaddressed risk factors: While factors such as ASA score and T classification are discussed, other important risk factors, such as alcohol consumption, albumin or hemoglobin levels, and intraoperative blood loss, were not included, even though they have been highlighted in the literature as relevant for the development of SSI.
Single-center data: The fact that the study was conducted at a single center limits the generalizability of the results. Differences in surgical or prophylactic practices between institutions were not explored, which could have strengthened the findings.

3. Discussion:
The discussion is aligned with the results in terms of identifying risk factors and the microbiology of SSI. However, it overlooks a more critical discussion of the study’s design and methodological limitations. The authors do not sufficiently address the difference between a cross-sectional study and a cohort study, which impacts how the conclusions should be interpreted.

Lack of discussion about design: The authors do not explicitly acknowledge that this is not a longitudinal cohort study and do not discuss the limitations this implies, such as the inability to properly analyze time-to-event data using Cox regression.
Limited comparison with previous studies: While previous studies are mentioned, the discussion could benefit from greater depth regarding how the results align or contrast with existing literature. This would provide a more robust view of the clinical and scientific implications of the findings.

4. Conclusions:
The conclusions are generally aligned with the results, but they lack important nuances. Although the authors highlight the low incidence of SSI and the identified risk factors, they do not fully acknowledge the study’s limitations. Additionally, the recommendations for clinical practice and future research are vague and could be more specific.

Insufficiently nuanced conclusions: The conclusions do not adequately address the methodological limitations of the study, such as the fact that this is a cross-sectional analysis rather than a cohort study. This could lead to misinterpretations regarding the predictive capacity of the results.
General clinical recommendations: The recommendations are useful, but they lack specific details about how the results could influence current clinical practice. It would be helpful if the authors offered clearer guidelines on how to implement infection control measures based on the identified risk factors.

5. Recommendations for Improvement:

Redefine the study design: The authors should reconsider the description of the study design and acknowledge that it is a cross-sectional study. This also implies that a more appropriate statistical approach would be necessary if it were indeed a cohort study.

Include a more comprehensive analysis of risk factors: Future studies should include analyses of additional factors such as alcohol consumption, preoperative albumin and hemoglobin levels, and intraoperative blood loss, which may have influenced the development of SSI.

Enhance the discussion of limitations: It would be important for the authors to delve deeper into the study's limitations, including the cross-sectional nature of the data, the lack of time-to-event analysis, and the restrictions on generalizability due to it being a single-center study.

Strengthen clinical implications: The conclusions should include more detailed recommendations on the implementation of infection control strategies based on the study's findings, with a clearer focus on how these results can be used to improve clinical practice.

Final Assessment: The study provides useful data on surgical site infections in laryngeal surgeries, but its value is limited by design issues and the lack of an appropriate time-to-event analysis. A significant revision of the study design and the inclusion of more robust statistical analysis in future work are recommended to improve the validity of the findings.

Author Response

(The authors gave the same response as above.)

Round 2

Reviewer 3 Report

Comments and Suggestions for Authors

Thank you very much for your answers.

The extensive explanation regarding the design and the use of the statistical models used in this study seems correct to me. I believe that the phrase introduced in the methodology regarding the design allows readers to perfectly understand the design and the monitoring of the cases under study. Personally, in the first review, it was difficult for me to fully understand the design. I think that the clarification introduced clarifies this perfectly.

As for everything else, your answers and changes in the text seem appropriate to me, and I think that the text has improved in quality.

Thank you very much